# Process Evaluation of Scandium Production and Its Environmental Impact

**Aratrika Ghosh [1], Soniya Dhiman [1], Anirudh Gupta [1] and Rohan Jain [2,\*]**

1  Department of Biochemical Engineering and Biotechnology, Indian Institute of Technology-Delhi, New Delhi 110016, India
2  Helmholtz-Zentrum Dresden-Rossendorf, Helmholtz Institute Freiberg for Resource Technology, 01328 Dresden, Germany
\*  Correspondence: rohanjain.iitd@gmail.com; Tel.: +49-351-260-2725

**Abstract:** With the advancement of technology and a global shift towards clean energy, the need for rare earth metals is increasing. Scandium, a rare earth metal, has been extensively used over the decades in solid oxide fuel cells and aluminum–scandium alloys that have a vast, evolving market in aerospace, automobiles and 3D printing. However, the market struggles to maintain the supply chain due to expensive production processes and the absence of uniform global distribution of primary sources. Therefore, identification of alternative sources and technological advancements for scandium recovery are needed. To this end, an effort has been made to provide a review of the advances in different technologies applied in scandium recovery from diverse sources. Emphasis has been given to the improvements and upgrades to technologies in terms of environmental impact and recovery efficacy. An attempt has been made to discuss and deliver a clear representation of the challenges associated with every source for scandium recovery and the major developments in solving them. The environmental impact of scandium recovery and recycling has also been discussed.

**Keywords:** hydrometallurgy; leaching; adsorption; solvent extraction; crystallization; life cycle assessment

## 1. Introduction

The release of carbon from the waste materials generated by the mining of minerals significantly influences the environment and the climate, which have a great impact on human health and biodiversity [1]. However, the role of mining is also important for the recovery of critical metals like gallium, indium, scandium and other rare earths that are used in high-tech and low carbon emission technologies [2]. Furthermore, the mining industries are also important from an economical point of view for various states and provinces where mining activities are mostly concentrated. Usually, the wastes generated during mining activities have been discarded as such in the form of slag and tailings dams, even though these contain valuable minerals. In addition, the failure of the tailings dams has led to unmitigated disasters in Brazil, Hungary and Italy [3]. Therefore, mining waste recycling may stimulate innovative local industries, reduce waste production and natural resources intake, prevent environmental damage and create financial assets. The assessment of environmental benefits after mining waste reprocessing and final disposal of waste is important. It is also necessary to employ cost and benefits methods to assess the expenditure of waste reprocessing and the monetary value of recovered metals. For this, a detailed understanding of the metal extraction process is important. In this review, scandium has been discussed as it is considered an expensive as well as rare metal due to its difficult extraction and poor distribution [4]. It is generated as an ore-processing by-product and is generally present in the waste liquors, tailings, slags and residues. However, since the concentration of scandium is very low in comparison to contaminating metals, its recovery becomes challenging, expensive and environmentally hazardous. Additionally, a

low recycling rate, irreplaceable application in green technologies, limited mining regions and high economic importance makes it therefore important to focus on high efficiency recovery. It is also referred to as a critical metal by Brazil, the USA and the European Union [5,6].

Scandium is one of the most highly abundant metals in the earth's crust, yet, it is not readily available for mining and extraction due to its dispersed nature. Scandium as an ore with a concentration greater than 40% has been reported only in Norway [7]. Other countries with main scandium resources are China, Russia, the United States of America, Madagascar, Australia and Kazakhstan [8–10]. Scandium predominantly occurs in traces along with other minerals in ores of cobalt, iron, nickel, tin, etc. [11,12]. The major application of scandium is as an alloy with metals like aluminum, magnesium, zirconium, etc. These alloys find extensive use in sports, military [9] aircraft industries [13] and solid oxide fuel cells (SOFCs), respectively [14]. At present, the worldwide market for scandium has a demand of 98 t/annum of scandium for aircraft and vehicles, which will increase to 3000 t/annum of scandium by 2032 [15]. Owing to this steep increase in demand, scandium has been classified as a critical metal [16].

With such a predicted rise in the demand of scandium in the upcoming decade, the production rate does not seem to increase accordingly. The low production and demand of scandium is a chicken and egg problem. The high price and limited market availability seem to discourage industries from using this raw material. Hence, the number of buyers is low. Due to the small market, metal producers are deterred from investing in the recovery of scandium, despite its high price. Presently, the amount of scandium produced is clearly insufficient to address the increasing demand across the world. Therefore, new supply strategies for scandium are highly desirable. Paradoxically, although scandium is abundant in the earth's crust, it is not an ore-forming metal. Therefore, it rarely occurs in concentrations above 100 ppm as a primary source, making its extraction economically unattractive [15]. Hence, new sources and recovery processes are required to break this cycle.

Few review articles have been published on scandium separation and recovery from different sources [12,17–20]. Wang and Cheng [12] reviewed the separation and purification of scandium by synergistic solvent extraction and solvent extraction with chelating extractants, focusing on their recovery performance as well as their extraction mechanisms. In addition, ion exchange and liquid membrane extraction for scandium recovery were also discussed. Zou et al. [17] reviewed the extraction chemistry of scandium in different extraction systems in detail. Wang et al. [18] reviewed scandium recovery through metallurgical pathway from various resources, including scandium ores, residues, tailings and waste liquors, focusing on the selection of processes to recover scandium as a minor element and incorporating the scandium recovery process into the main flowsheet for the production of the main metal. Junior et al. [19] also reviewed scandium extraction from different sources. However, the primary focus was on eco-friendly processing and clean technologies. Pyrzynska et al. [20] discussed some recent progress made in the research on scandium separation, purification and pre-concentration from different sources such as ores, electronic waste (e-waste), water, sediment, soil and plants both in industrial as well as laboratory systems. However, consolidated information on the advances in the recovery of scandium from various sources, especially from mining tailings/e-waste is not discussed properly. Moreover, the diversity in the treatment process and the associated prospects and consequence of these methodologies with respect to their environmental impact and metallurgical gain have not been discussed properly. Different metallurgical operations have been used to recover scandium from secondary sources. One such choice is pyrometallurgy, which is a conventional method to recover different metals from different type of wastes. However, high energy consumption and generation of toxic substances are the disadvantages associated with pyrometallurgy [21]. In recent times, the principal technologies being used to recover scandium from secondary sources are based on hydrometallurgical processes. Hydrometallurgical processes are suitable alternatives for the

treatment of industrial materials waste because of their cost effectiveness, reduced formation of poisonous gases and dust, operational selectivity, simple control of the procedure and effectiveness [22,23]. In comparison to pyrometallurgy, hydrometallurgical processes utilize low temperatures and are more predictable and controllable, which make them more environmentally friendly. Therefore, the main objective of the present review is to give a clear portrait of the growth in hydrometallurgical processes regarding scandium recovery from various sources especially mining tailings/e-waste. It will provide a deep insight into the wide-ranging technologies that are available, from leaching to the use of polymer inclusion membranes for the recovery of the metals, along with their advantages and disadvantages. Further, to support the idea, discussions on several cases of scandium recovery carried out across the world for the last decade have also been included.

## 2. Scandium Sources

Scandium seldom occurs in concentrated quantities and is usually found distributed as traces in rocks containing ferromagnesium reserves [18]. Worldwide, there is about 2 million tons of scandium, with China accounting for 27.5% of the total reserves [24]. However, most of the reserves cannot be exploited due to technical, economic and environmental challenges [19]. Therefore, the extraction of scandium in pure form is complicated as well as expensive [24]. Across the globe, scandium occurs in more than 800 variants of minerals in low concentrations and a highly complexed state. The scandium minerals containing considerable quantities of the metal (i.e., more than 20 wt%) are thortveitite ($Sc_2Si_2O_7$), pretulite ($ScPO_4$) and kolbeckite ($ScPO_4 \cdot 2H_2O$) [25].

### 2.1. Scandium Abundance

Major scandium ore deposits have been found in the USA, Norway, Australia, China, Russia, Madagascar, Kazakhstan and Ukraine [8,11]. Any resources with scandium concentration between 20–50 mg/kg can be considered as ore for exploitation [26]. In USA and Kazakhstan, scandium sources are mainly ores of aluminum uranium, zirconium and tantalum. In Australia, it occurs in ores of nickel laterite. In China, Russia and Ukraine, scandium is found in ores of tungsten, iron and tin, while in Madagascar and Norway, it occurs in pegmatite rocks. Globally, the principal source of scandium is niobium-rare earth element-iron (Nb–REE–Fe), the largest REE resource and second largest resource of scandium in the world. It is located in Inner Mongolia, China and accounts for approximately 90% of global scandium production. In Bayan Obo, scandium is regenerated as a by-product of mining of the other REEs and iron [27]. The scandium content of the Bayan Obo deposit ranges between 26–110 ppm in various ores and it reaches 163 ppm of scandium in the REE ore tailings [28]. Scandium in the Bayan Obo deposit is mainly hosted by Aegirine. Numerous lateritic deposits in eastern Australia (New South Wales and Queensland) that have sumptuous scandium content and are being considered for viable mining [29]. A mining lease has been awarded for the Nyngan deposit in New South Wales, Australia. The laterites in these deposits are generated due to intense weather conditioning of ultramafic and mafic rocks. These rocks have concentrated scandium in the range of 100–400 ppm by the adsorption of geothite or incorporation into the hematite structure [29]. The Zhovti body deposit in Russia is graded at 105 ppm scandium and contains aegirine as its main scandium bearing mineral (Table 1) [25]. The most important and largest source of scandium in Russia is the Kovdor baddeleyite–magnetite–apatite deposit. It contains a scandium reserve of 420 tons and the grade of the ore is 800 ppm of scandium [30]. The other major source of scandium in Russia is Tomtor, one of the world's largest carbonatites contributors, which has elevated concentrations of the REEs, including scandium up to 570 ppm (Table 1) [31–33]. High scandium content (45%) is found in thortveitite-rich pegmatites in Madagascar and Norway. It is also reported that in the Iveland-Evje district of Norway, scandium-bearing pegmatites contain approximately 1000 ppm scandium [25]. Table 1 shows some of other primary sources of scandium.

**Table 1.** Concentration of scandium in different mineral ores other than bauxite ores.

| S. No. | Mineral Ore | Scandium Content | References |
|--------|-------------|------------------|------------|
| 1. | Aegirine (Russia) | 105 ppm | [25] |
| 2. | Pegmatites | 1000 ppm | [25] |
| 3. | Aegirine (Bayan Obo) | 26–110 ppm | [28] |
| 4. | Lateritic deposits | 100–400 ppm | [29] |
| 5. | Baddeleyite–magnetite–apatite | 800 ppm | [30] |
| 6. | Araxa (SE Brazil) complex REE (Nb–P) ore | 219–322 ppm ($Sc_2O_3$) | [32,33] |
| 7. | Tomtor deposit | 570 ppm ($Sc_2O_3$) | [32,33] |
| 8. | REE–monazite | 15 ppm | [34] |
| 9. | REE–allanite | 24 ppm | [35] |

*2.2. Scandium from Secondary Sources (Mining Process and End-of-Life Products)*

Scandium was classified as a critical metal owing to its steep increase in demand [16]. An economically profitable option is to recover scandium from waste streams of scandium-associated ore processing units. Scandium is found in a large extent in aluminum ores. Bauxite is the most common aluminum ore, which has a high content of scandium. The Bayer process associated with aluminum extraction results in enrichment of the scandium concentration in the bauxite residue [36]. The concentration of rare earth elements in bauxite varies from their region of origin. Table 2 gives the scandium concentration in red mud from studies done in recent years for its recovery. A review by Zang et al. gives a detailed discussion about the sources, concentration and scandium recovery from red mud [10]. In red mud, the approximate concentration of rare earths varies in the range of 500–1700 ppm, with scandium concentrations ranging between 130–390 ppm [37]. If all the scandium present in red mud could be recovered, then 6600–20,400 t/annum would be available [15]. Several rare earth metals are present in bauxite residue. However, the maximum concentration in terms of presence belongs to scandium which is around 95%compared to rest rare earth metals present in the waste [38]. However, the presence of other metals like iron, aluminum, etc., in large concentrations hinders the hydrometallurgical processes making it difficult to recover such high concentrations.

**Table 2.** List of countries reported to be generating bauxite ore in recent years and the scandium content of the ore.

| S. No. | Country | Scandium Content | Reference |
|--------|---------|------------------|-----------|
| 1. | Jamaica | 550 ppm | [15] |
| 2. | Canada | 31,100 ppm | [39] |
| 3. | Greece | 20 ppm | [40] |
| 4. | China | 20–38 ppm | [41] |
| 5. | Russia | 70–120 ppm | [42] |
| 6. | Germany | 57 ppm | [42] |
| 7. | Hungary | 94 ppm | [42] |

One important source of scandium is coal/coal combustion byproducts (CCPs) [43–49]. CCPs accumulate at an annual rate of 115 million tons in the United States alone and contain an average scandium concentration of 36–70 ppm along with up to 1000 ppm rare earth metals (REEs). Technoeconomic analyses suggest that scandium extraction from these waste residues will be critical for industrially viable recovery of REEs since scandium comprises more than 90% of the total REE value at current prices [46]. Scandium at low concentrations is also found with transition and rare earth elements like molybdenum, tungsten, titanium, tantalum, yttrium, zirconium, etc. [50]. The rare earth minerals such as bastnasite and monazite contain scandium in the range of 20–50 ppm [51]. Scandium is also found in ores of uranium as uraninite in Russia, Kazakhstan, Australia, Namibia

and Canada [52]. It is also recovered from titanium ore like magnetovana–ilmenite located in Panzhihua, China at a concentration as high as 0.04% [53]. A report by the United States Geological Survey, (Mineral commodity summaries, 2019) stated that approximately 5,400,000 metric tons of ilmenite ($FeTiO_3$) and 750,000 metric tons of rutile ($TiO_2$) are mined across the globe annually [54]. Apart from that, major titanium ores are found in India, Norway, Australia, Canada and South Africa [16], accounting for a potential global yield of scandium ($Sc_2O_3$) in the range of 96–194 tons. A noticeable amount of scandium is also found in tungsten [55], nickel and cobalt ores. Nickel laterite ores from Cuba, the Dominican Republic and New Caledonia [56] have high concentration of scandium, up to 100 ppm, and are a probable source for exploitation. Apart from the sources mentioned above, there are many new potential sources that have been identified, such as electronics and electrical waste, municipality waste, coal fly ash, phosphor-gypsum and phosphate rocks and many more that have been identified to contain recoverable amounts of scandium [19]. The main problem with the exploitation of these sources is the irregularity in the metal's concentration. For example, use of e-waste as a scandium source needs collection and segregation of the waste, which is an additional challenge. Moreover, the exploitation of the phosphorus reserves as a scandium source is restricted due to their insufficiency [53].

## 3. Scandium Applications

The two main applications of scandium are in solid oxide fuel cells and aluminum–scandium alloys. An enhanced oxygen ion conductivity can be attained in solid oxide fuel cells through scandium doping with zirconia in place of yttrium [19]. Aluminum–scandium alloys are among the most promising candidates for light weight alloys [57]. The advantages of a minor addition, generally 0.2–0.6 wt% of scandium in aluminum alloys, are improved strength, thermal resistance, weldability and corrosion resistance. The main application of aluminum–scandium alloys is in the aerospace and automotive industries and for sports equipment (bikes, baseball bats, etc.) which rely on high performance materials. Scandium, along with aluminum and magnesium as an alloy is used in 3D printing which finds application in the aerospace and automobile industries. $Sc_2O_3$ is used in Erbium-doped yttrium–aluminum–garnet (Er:YSG) crystals for optics in laser applications [37].

Due to its specific mechanical and chemical properties, the applications of scandium are increasing, which has heightened the market demand for scandium. However, it is difficult to extract scandium because it is sparsely distributed in trace amounts in natural minerals, and the resources of scandium mineral deposits usually exist in complicated forms. There is an urgent need to separate and recover scandium from secondary resources.

## 4. Possible Flow Sheets for Scandium Recovery

A cascade of different operations is generally required to obtain and enhance the purity of any metal recovered from its ore or from byproducts of other metal extraction processes. The principal steps from the mining to the extraction of its purest form include liberation, separation/upgrading and purification. The most commonly used processes for metal processing for minor elements in hydrometallurgy include leaching, solvent extraction, ion-exchange, precipitation and electrochemical refining. The choices of the processes are based on the type of ore being handled. However, different options were developed in addition to the classical approach [58]. In the classical approach, scandium was recovered from uranium ores in Russia. The process began with low-concentration sulfuric acid leaching, followed by solvent extraction with 0.1 M dodecyl phosphoric acid solvent in kerosene to extract uranium. This led to the build-up of concentrated scandium, titanium and thorium in the organic phase leading to solvent poisoning. Stripping of the organic solvent with HF acid led to the precipitation of a scandium–thorium fluoride cake. The cake was then double-digested with NaOH for conversion to scandium hydroxide. The crude scandium hydroxide was digested again with hydrochloric acid to remove impurities like thorium, titanium, uranium, etc. Once the precipitate was removed, the filtrate was further

treated with controlled oxalic acid to recover scandium oxalate precipitate. The scandium oxalate cake was then calcined below 800 °C to generate scandium oxide. To enhance the purity of the oxide obtained, it was further dissolved in hydrochloric acid and stripped with ammonia. The scandium oxide formed was again calcined at 700 °C to generate scandium oxide with a purity of 99.5%. This process applies reagents, which are corrosive as well as poisonous resulting in the generation and release of toxic byproducts. Hence, it is clearly understandable that this process contributes to the environmental burden as it utilizes large amounts of hazardous chemicals and generates toxic products. However, since no life cycle assessment (LCA) has been done on the recovery of scandium following the classical approach from uranium ores, a direct relationship with indicators cannot be concluded. Recently, a study done by Wang et al. [27] discussed the environmental impact of scandium production from the Bayan Obo Mines. Figure 1 shows the processes followed in the recovery of Sc$_2$O$_3$ from the Byan Obo Mine's REEs tailings. The process is studied from ore mining to the final scandium products. It includes primary, secondary and tertiary separation of iron, other REEs, scandium and niobium. Once other major impurities are separated, scandium is further concentrated and purified by processes like pressure filtration, pressure acid leaching, extraction, back extraction, calcination, precipitation and many more. The environmental impact for the production of 1 kg of Sc$_2$O$_3$ was analyzed with the help of the Tools for Reduction and Assessment of Chemicals and Other Environmental Impacts (TRACI 2.1) which include ten potential environmental indicators some of which are Ecotoxicity, Human Toxicity Cancer, Human Toxicity Non-Cancer, Global Warming Air (GWA) and many more. From the assessment, it was seen that the process has a major environmental burden in form of GWA, with extraction and separation being the major contributors. However, in the present case where scandium contributes only 0.01% of the ore, the removal of the other REEs and iron takes up a large amount of energy as well as material. Hence, along with identification of new potential sources of scandium, advancement of technology that can reduce this burden is very important. As time advanced, new methodologies were developed and proposed to overcome the environmental problems along with the ability to recover the maximum scandium available in the source. The following sections will discuss all the possible alternatives that have been identified for scandium recovery.

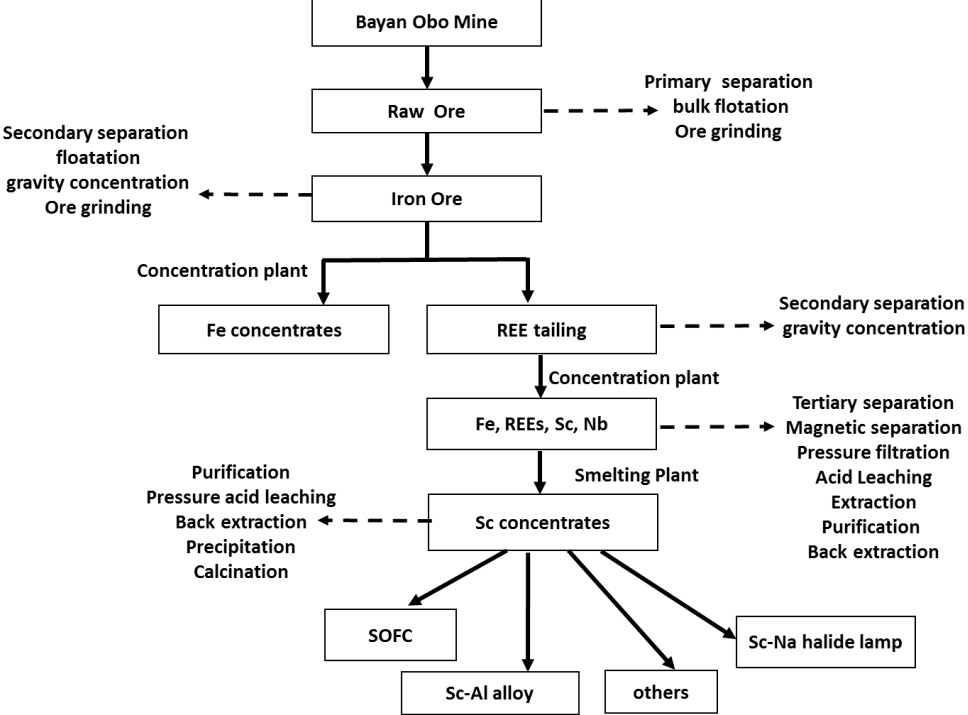

**Figure 1.** Sc$_2$O$_3$ from Byan Obo Mine's REEs tailings.

## 5. Separation Processes for Scandium Recovery

As scandium occurs in concentrations around 0.002–0.005% in ores of other metals, the recovery processes become complex as well as expensive. The major processes employed for recovery of scandium are hydrometallurgical in nature, such as extraction, adsorption filtration, ion-exchange, solvent extraction, etc. The flow of these processes is dependent upon the initial source and most of the process begins with leaching followed by a series of different hydrometallurgical processes. Thus, scandium's rare availability coupled with the complex recovery processes adds to its cost of production making it highly expensive. According to the US Geological Survey report [49], the price of 99.99% pure $Sc_2O_3$ is USD 2200 per kg and the global supply and consumption of scandium oxide was estimated to be about 15–25 tons per year [59]. In spite of the increasing cost, scandium demand is increasing for modern technologies in automotive, optical, electronics and related industries [9], as discussed above. Thus, different processes for the recovery and purification of scandium are in urgent need.

### 5.1. Chemical Leaching

Chemical leaching is one of the oldest and most important operations in metallurgical industries for the extraction of metals from their ores. It is a process through which the metals trapped in ore and industrial process waste are recovered using strong mineral acids. The performance of different leaching systems from different secondary sources reported in the literature for scandium leaching is reviewed and tabulated in Table 3. Li et al. [60], Wei et al. [61] and Xiao et al. [24] studied the recovery of scandium using hydrochloride acid (HCl) as the leaching agent. They further compared their studies with other mineral acids and concluded that the scandium leaching rate was either high or similar to the other leaching agents. Ochsenkuehn-Petropoulou et al. [62] and Rivera et al. [63] used $H_2SO_4$ as the leaching agent for the scandium leaching from bauxite residue, but it showed poor leaching efficiency among all the tested systems. Furthermore, Bonomi et al. [64] studied scandium recovery using ionic liquid at 200 °C. The biggest hurdle faced in the direct leaching of bauxite residue is the development of silica gel. Alkan et al. [65] provided a solution by preventing the synthesis of silica gel using hydrogen peroxide ($H_2O_2$). However, it affected the recovery as it reduced the leaching efficiency. Thus, as observed, the use of $H_2O_2$ can avoid silicon leaching and consequent silica gel formation.

**Table 3.** A brief summary of different systems of scandium leaching from secondary sources.

| S. No. | Real Sample | Leaching System | Leaching Conditions | Leaching Rate of Scandium (%) | Reference |
|---|---|---|---|---|---|
| 1. | Scandium Rough Concentrate | HCl | s/l = 1/1.5, 60 °C, 90 min | 95.1 | [24] |
| 2. | Red Mud | HCl | 75 °C, 2 h | 99.97 | [60] |
| 3. | Red Mud | HCl | s/l = 1/10, 80 °C, 3 h | 83.9 | [61] |
| 4. | Bauxite Residue | $H_2SO_4$ | s/l = 1/5, 90 °C, 60 min. | 50 | [62] |
| 5. | Bauxite Residue | $H_2SO_4$ | s/l = 1/20, 25 °C, 24 h | 40 | [63] |
| 6. | Bauxite Residue | 1-ethyl-3methyl imidazolium hydrogen sulphate | s/l = 5% $w/v$, 200 °C, 12 h | 80 | [64] |
| 7. | Bauxite Residue | $H_2SO_4 + H_2O_2$ | s/l = 1/10, 90 °C, 30 min | 68 | [65] |
| 8. | Red Mud | HCl + $H_2O$: Red mud: EDTA | 40 mL HCl, 10 g red mud, 2 g EDTA, 70 °C, 4 h | 79.6 | [66] |
| 9. | Bauxite Residue | $CO_2 + H_2SO_4$ | s/l = 1/3, 30 °C, 6 h | 50 | [67] |
| 10. | Bauxite Residue | $H_2SO_4$ | s/l = 1/50, 80 °C, 60 min | 60 | [68] |
| 11. | Fe-Ti Residue | $H_2SO_4$ | s/l = 1/7, 95 °C, 5 h | 85–95 | [69] |
| 12. | REE Silicate | $H_2SO_4$ | s/l = 1/30, 200 °C, 15 h | - | [70] |
| 13. | Blast Furnace Slag | $H_2SO_4 + H_2O$ | 400 rpm, 200 °C, 10 min | 83 | [71] |
| 14. | Bayan Obo Tailings | $H_2SO_4$ | s/l = 1/4, 245 °C | 96 | [72] |
| 15. | Nb Ore Concentrate | HCl | s/l = 2.2, 100 °C | 97 | [73] |

*5.2. Bioleaching*

Bioleaching, also known as microbial leaching or biomining, is a process where metals are solubilized from insoluble solid matter either by direct metabolism by the microbes or by the metabolism products of the microbes [74]. This procedure helps in recovering valuable metals from low-grade ores, industrial process wastes, mine tailings, etc., in a more effective and environment friendly way. This phenomenon typically occurs as a result of the microbe generating organic or inorganic acids in their metabolic pathway [75].

Bioleaching can be done either by using heterotrophic or autotrophic microorganisms. The autotrophic process follows bio-oxidation/reduction cycles in the presence of inorganic acids, while heterotrophic leaching is carried out by the produced organic metabolites that complexes with the metal leading to the formation of soluble chelates. Heterotrophic microorganisms have successfully leached scandium from primarily bauxite and fly ash residues (Table 4). Most of the microbes studied for the leaching of scandium produced mainly oxalic acid. The high production of oxalic acid is promoted by the high pH (6–10) of the system [76]. Apart from oxalic acid, other organic acids produced by the microbes are acetic acid, succinic acid, gluconic acid, malic acid, citric acid and lactic acid. In all the bioleaching studies mentioned here, the system follows low pulp density and one-step bioleaching. The stimulating effect of interactions between strains and scandium are long-lasting in one-step processes. Moreover, increasing the pulp density causes loss of microbial metabolic activities, inhibiting growth and resulting in poor leaching of the metal [77]. However, exceptionally high leaching efficiency (94%) has been seen when leaching has been done using *Gluconobacter oxydans* from red mud [78]. Only two studies have been reported that showed leaching of scandium from red mud through gluconic acid produced by *Gluconobacter oxydans* [79,80]. However, no such detailed discussion on the mechanism has been reported. Therefore, further detailed studies with gluconic acid on other scandium sources can be carried out in the future.

**Table 4.** A brief summary of different systems of scandium leaching from secondary sources.

| S. No | Sample | Leaching System | Leaching Conditions | Leaching Agents | Leaching Rate of Scandium (%) | Mechanism | Reference |
|---|---|---|---|---|---|---|---|
| 1. | Red Mud Indian and German | chemoorganotrophic microorganisms, *Gluconobacter oxydans* (DSMZ 46616) | 10% pulp density, 37 °C, was observed after 18–20 d, 120 rpm | gluconic acid | 83% and 94%, respectively | - | [80] |
| 2. | Red Mud | *Penicillium tricolor* (RM-10) | 10 days, 2% pulp density, one-step bioleaching | citric, oxalic, and gluconic acids | 70% | Detoxification | [79] |
| 3. | Residual Fly Ash | *C. bombicola, C. curvatus, P. chrysosporium* | Fly ash leached with supernatant 28 °C, 6 h, 50 rpm, 1% pulp density | - | 63, 48.5 and 52.1, respectively | - | [81] |
| 4. | Ash–Slag Waste | acidophilic chemolithotrophic microbial communities | 45 °C, 10 days, 10% pulp density, pH 2.0 by adding sulfuric acid | sulfuric acid | 52% | - | [82] |
| 5. | Red Mud | chemoheterotrophic bacteria, *Acetobacter sp.* | 30 °C, 120 rpm, 2% pulp density, one-step | succinic acid acetic acid, malic acid, oxalic acid, lactic acid | 52% | Acidolysis, complexolysis | [83] |
| 6. | Red Mud | fungal strain *Aspergillus niger* isolated from pistachio husk and grape skin | 30 °C, 150 rpm, 20 days, 3% pulp density | citric and oxalic acids | 29% and 38%, respectively | Detoxification and complexation by acidic metabolites | [84] |
| 7. | Bauxite Residue | *Acetobacter tropicalis* | one-step bioleaching process at 1% s/l, 30 °C, 120 rpm, 20 days | acetic, oxalic, and citric acids | 42% | Detoxification and complexation | [85] |

Compared to conventional leaching methods, bioleaching has higher selectivity towards scandium in the presence of impurities. Moreover, it is a greener and better option for the extraction of metal from solid materials as it has operational flexibility, low cost, no toxic by-product generation, less energy consumption and is environmentally friendly [79]. The most significant disadvantage that has been observed is the low leaching efficiency

and high leaching time compared to conventional mineral acid leaching. Bioleaching is a significantly less exploited process for extraction of REEs. It has a huge scope in exploring scandium recovery from other sources, such as waste printing circuit boards, which contains neodymium, dysprosium, lanthanum, scandium and yttrium along with zinc, nickel, copper and gold. Bacterial or fungal bioleaching of scandium as well as rare earth elements from waste printing circuit boards, has not been addressed properly [78].

## 6. Recovery Processes

### 6.1. Liquid/Liquid Extraction

Liquid/liquid extraction, also known as solvent extraction, is a traditional method used extensively in the chemical industry for the separation of compounds based on their relative solubility in two different immiscible liquids [41]. For scandium recovery from various processing liquid wastes and leachates, solvent extraction is widely used because it generally offers the advantages of good processing capacity, operational ease at larger scales and lower operating costs. Different types of extractants, such as acidic, basic, neutral, chelating, as well as synergetic extraction systems have been explored for the extraction and purification of scandium [86]. The extraction studies that have been reported in the literature is reviewed and summarized in Table 5. These studies suggest that acidic organophosphorus extractants such as di-(2-ethylhexyl)phosphoric acid (DEHPA) and bis(2,4,4-trimethylpentyl)phosphonic acid (Cyanex 272), were the most applied extractants for the separation of scandium from various other metal ions. Due to its high charge density, scandium generally shows a higher extractability than the other rare earths. Upon application of Cyanex 272 and 923 (mixture of four tri-alkylphosphine oxides) in sulfuric acid media, the recovery of scandium was >98%, whereas selective separation of scandium from thorium and zirconium was achieved using Cyanex 572 (organo-phosphorus containing phosphinic and phosphonic acids) in HCl media. Selective separation of scandium from Fe was reported using betainium bis(trifluoromethanesulfonyl)imide ([Hbet][TF$_2$N]) in aqueous medium by Onghena et al. [87]. The percentage extraction of scandium was 99% using D2EHPA + tributyl phosphate (TBP), while 90% scandium was stripped when D2EHPA + primary amine (N1923) was applied as the extractant. The extraction percentage was greater than 99% using 2-ethylhexyl phosphonic acid mono-2ethylhexyl ester (P507) + TBP and trialkylphosphine oxide (TRPO) as extractants in different media. It is evident from the given data that sulfuric acid media was widely used for the recovery of scandium. The extraction percentage is greater than 90% using most of the reported extractants, which suggests that application of these extractants for the recovery of scandium is feasible. The comparative data also indicate that in order to obtain highly pure (>99%) scandium oxide, P507 + TBP and TRPO can be applied as extractants.

**Table 5.** Summary sheet on the extraction of scandium using different extractants.

| S. No. | Starting Metals (mg/L) | Extractant | Aqueous Medium | Extraction Mechanism | Extracted Metal Ions | Comments | Reference |
|---|---|---|---|---|---|---|---|
| 1. | Sc (9.9), Th (8.9), Ti (30.7), Zr (1.3), Fe (13,091.4), Mn (9530.9), REs (40.5), Al (506.5), Ca (5591.9), Mg (221.1) | Cyanex 572 | HCl | $M^{+3}_{(aq)} + 3HL_{(org)} = ML_{3(org)} + 3H^+$ | Sc, Th, Zr | Selective separation of Sc from Th and Zr; HCl as stripping agent | [55] |
| 2. | Sc (1.8 mol/kg) | [Hbet][TF$_2$N] | - | $M^{+3} + 3[Hbet][TF_2N]_{(org)} = [M(bet)_3(TF_2N)_{3(org)} + 3H^+$ | Sc, Fe | Selective separation of Sc from Fe using scrubbing; HCl as stripping agent | [88] |
| 3. | Sc (139) | Cyanex 272+ Cyanex 923 | H$_2$SO$_4$ | $Sc^{+3} + (HL)_{2(org)} + B_{(org)} = Sc(HL_2)B(SO_4)_{org} + H^+$ | Sc | 98.79% Sc is recovered using oxalic acid as stripping agent | [89] |
| 4. | Sc (9), Fe (22), Al (203), Si (28), Na (5837), Ca (416) | P204 P507 Versatic 10 | H$_2$SO$_4$ | $M^{+3} + 3(HA)_{2(org)} = MA_3.3HA_{(org)} + 3H^+$ | Sc, Fe | P204 is a better extractant than P204 and Versatic acid 10; 97% recovery of Sc | [90] |
| 5. | Sc (23.6) | P507 + isooctanol | H$_2$SO$_4$ | $Sc^{+3} + 3(HA)_{2(org)} = Sc(HA_2)_{3(org)} + 3H^+$ | Sc, Zr, Ti | $SF_{(Sc/Zr)} = 34$, $SF_{(Sc/Ti)} = 494$; 99% Sc is recovered using H$_2$SO$_4$ as stripping agent | [91] |
| 6. | Sc (4.33), Na (23,800), Fe (107), La (14.4), Ti (0.08), Ca (400Al (2510), Y (15.3), Ce (30.3), Nd (3.06), Dy (1.74) | [Hbet][TF$_2$N] | H$_2$O | $M^{+3} + 3[Hbet][TF_2N]_{(org)} = [M(bet)_3(TF_2N)_{3(org)} + 3H^+$ | Sc, Fe | Separation of Sc from Fe is achieved by reducing Fe; ascorbic acid as reducing agent; H$_2$SO$_4$ as stripping agent | [88] |
| 7. | Sc (5.53), Ca (611), Fe (1653), Ti (311), V (49.1), Cr (9.36), Zr (5.91), Ga (2.00) | D2EHPA + TBP | H$_2$SO$_4$ | - | Sc | D2EHPA is selective extractant for Sc; %E = 99%; TBP used as phase modifier; Sc is recovered as Sc(OH)$_3$ using NaOH as stripping agent | [92] |
| 8. | Sc (365), Ti (579), Fe (6), Zr (53.9) | TRPO | H$_2$SO$_4$ + H$_2$O$_2$ | $Sc^{+3}_{(aq)} + HSO_4^-{}_{(aq)} + SO_4^{2-}{}_{(aq)} = HSc(SO_4)_{2(aq)}$ $HSc(SO_4)_{2(aq)} + 2TRPO_{(org)} = HSc(SO_4)_2.2TRPO_{(org)}$ | Sc | H$_2$O$_2$ is added to prevent the extraction of Ti; 99.9% stripping Sc using oxalic acid; 95% Sc$_2$O$_3$ is recovered with 99.34% purity | [93] |
| 9. | Sc (17), Ti (3875), Fe (5562), Al (8431), Ca (29), Na (4824), Mg (1521) | P507 + TBP | H$_2$SO$_4$ + CaF$_2$ | - | Sc | 99% pure Sc$_2$O$_3$ is recovered after stripping, precipitation and calcinations; phase modifier is required | [94] |
| 10. | Sc (23), Ti (2400), Fe (28,360), Mn (2400), Al (1030), Ca (1500), Mg (1900) | D2EHPA + N1923 | H$_2$SO$_4$ | $Sc^{+3} + (HL)_{2(org)} + [(RNH_3)_2(SO_4)]_{2(org)} + SO_4^{2-} = Sc(HL_2)[(RNH_3)_2(SO_4)]_2(SO_4)_{(org)} + H^+$ | Sc, Ti | 90% Sc is stripped using HNO$_3$; 80% Sc$_2$O$_3$ with 90% purity is obtained after precipitation and calcination | [95] |

*6.2. Adsorption*

Adsorption is a surface phenomenon where the transfer of molecules from the bulk fluid occurs on the solid particle. The transfer can be physical or chemical in nature; however, it is usually reversible in nature. The reverse of adsorption is desorption in which the molecule is released back, regenerating the adsorbent without altering the chemistry of the adsorbent. Table 6 discusses different kinds of adsorbents that have been used for the recovery of scandium such as ion exchange resins, solvent impregnated resins, bio-sorbents and many more. Carbon-based adsorbents like wood dust biochar, coconut shell activated carbon nanotubes and pristine were able to adsorb scandium selectively over rare earth elements like cerium and neodymium in the acidic media. Although wood dust biochar has shown low adsorption capacity compared to those available commercially, it is cheap and developed from agricultural waste giving it a sustainability and economic advantage. Sol-gel-processed silica doped with bifunctional ionic liquid trioctylmethylammonium 1-phenyl-3-methyl-4-benzoylpyrazol-5-onate [96] and $Fe_3O_4@SiO_2$ nanoparticles functionalized with the coupling agent (3-aminopropyl) triethoxysilane (APTES) and ethylenediamine tetraacetic acid (EDTA) as a ligand were also used as adsorbent with high selectivity towards scandium compared to other rare earth elements [97]. The latter is a magnetic adsorbent with an adsorption capacity as high as 95% as well as being easy to handle. However, nothing has been reported regarding the elution of the metal from the adsorbent which is important for the recovery of scandium. A recent study showed that zirconium phosphate (ZrP), when used as an adsorbent, showed high selectivity towards scandium compared to iron (III) with a separation factor of approximately 23. Around 99.9% pure scandium was recovered, although the elution after two cycles could reach only 60% [98]. Adsorption overcomes most of the drawbacks of solvent extraction, such as loss of solvent, generation of chemical sludge, multiple stage operation, emulsification, etc. However, it has its own drawbacks, such as slow processing, time consuming, low specificity towards selective ions and low efficiency desorption. In this review, other kinds of adsorbents used and their related problems, advantages and disadvantages will be discussed.

*6.3. Ion Exchange*

Ion exchange is a chemical treatment process where metal ions from the solvent are exchanged with ions on the resins. In this process, the ions in a solution are replaced by ions attached to a solid phase. Thus, the ions present in solutions are replaced by different ions originally present in the resin. These ions are of different types but of the same polarity. Ion exchange has been applied for recovery of scandium from as early as 1957, where it was recovered from a mixture of scandium, vanadium and titanium with the help of Dowex 1. The adsorbed scandium was recovered through desorption with 0.1 M oxalic acid and 0.1 M HCl [99]. Though the ion exchange technique is not new, it has undergone massive changes over time, just being based on the principal of reversible adsorption of ions at the solid/liquid interface. The developments were made in resin utilization and activity enhancement for maximum extraction. To date, various kind of resins has been employed for the recovery of scandium from different type of real industrially processed feeds like zirconium tailing waste, uranium leachate, copper leachate, coal ash leachate, red mud leachate, etc. which has scandium along with other REEs, and high concentration of impurities like thorium, iron, aluminum, etc. that strongly interfere with the recovery process. As most feed sources containing scandium are acidic in nature, cationic resins are the primary choice for recovery [100–105]. However, anionic resins are also been reported for the recovery of scandium [106,107]. From Table 6, most of the resins lack high selectivity towards scandium. Hence with scandium, other REEs as well as impurities, are adsorbed. While eluting, along with scandium, some impurities like aluminum [105], REEs [108], thorium [108] and iron [109] are desorbed as well. Eluting methods have been further developed and modified where selective elution of scandium in the presence of other REEs from the resin was achieved to some extent [107,108].

The biggest hurdle faced in the retrieval of scandium from leaching solution is the separation and segregation of scandium from other metals especially like iron due to their chemical resemblances. To overcome this issue, functionalized resin materials have been used [109–112]. Ethyleneglycol tetraacetic acid (EGTA)-functionalized chitosan–silica particles and 732 resins were able to selectively adsorb scandium over iron [110]. 732 resins were used in a two-step adsorption system where the resin had an affinity for scandium, iron and aluminum. In the first stage, eluent containing $Fe^{3+}$ is reduced in the presence of ascorbic acid and then scandium is complexed with EDTA. Since the resin had an affinity for iron ($Fe^{2+}$) and aluminum ($Al^{3+}$), they are adsorbed in the second stage adsorption, with only Sc–EDTA left in the eluent. The EGTA functionalized chitosan–silica possessed a much higher affinity for scandium over iron (by a factor of $10^5$) which is then selectively eluted at pH 0.5 with nitric acid. Thus, ion exchange resins can be considered as a workable separation technique for the reclamation of metals from dilute solutions. However, they may involve many operational challenges, such as damaged resin, fouling and high costs [12]. They are re-usable, durable, less complex, have low maintenance costs, and are easy to operate with adjustable selectivity [113,114]. However, elution is an important parameter as recovery of the metal is the primary goal, which is often compromised. Therefore, more attention is needed towards developing a complete, optimized process of adsorption and desorption.

### 6.4. Immobilized Extractants

This is a new technique consisting of the immobilization of liquid extractants on different substrates. It involves the exchange of ions with chelating agents or solvents impregnated on resins. It is a simple and effective tool to selectively separate metal ions from aqueous solutions and effluents. Chelating resins and solvent impregnated resins can show an explicit affinity for certain metals due to the functional group(s) assimilated to the support matrix that possesses the advantages of both extraction and adsorption [115]. For the solvent to work effectively, proper sorption of the extractant by the polymeric support is very important. This is generally achieved through hydrophobic interactions and, sometimes, partly through polar or electrostatic forces. Using solvent-impregnated resin, scandium can be recovered with high selectivity from low concentration solutions containing high amounts of impurities [115–117]. When treating feeds with scandium and other REEs, solvent-impregnated resin effectively adsorbed scandium, which was successfully eluted to recover scandium metal [118]. When treating feeds from bauxite- and laterite-processed ores with high concentrations of iron and aluminum, which heavily interferes with selective scandium adsorption, solvent-impregnated resins have shown selectivity in the order scandium > iron > aluminum [116,119]. However, the elution of scandium was difficult as most of the studies have either not reported about elution [115,119,120] or claimed it to be challenging and problematic [115]. However, in a study done with XAD-7HP resin impregnated with extractants 2-ethylhexyl phosphonic acid mono-2-ethylhexyl ester (PC-88A) and neodecanoic acid (Versatic 10), scandium could be selectively recovered in presence of impurities such as iron, aluminum, zinc, etc. without undergoing any scrubbing or reduction of iron ions. Scandium could also be substantially eluted from the system with 2 M sulfuric acid [117]. Solvent-impregnated resin is an alternative separation method that is environmentally friendly, faster and relatively easy to handle, having tunable selectivity towards any ion compared to ion exchange and solvent extraction methods alone. However, the disadvantages of the process are the loss of impregnated solvent over the course of time due to its solubility in the aqueous phase and the challenging elution process of the metal ion for successful recovery.

### 6.5. Biosorption

Biosorption is the process by which biological agents such as bacteria, fungi, etc. act as absorbents (bio-sorbents) to accumulate heavy metal in their cells either via metabolic pathways or physico-chemical methods. Amino, carboxyl, hydroxyl, sulfhydryl and phosphate groups of polysaccharides, lipids and glycoproteins present on the microbial surface serve as the binding site for metals [121]. Biosorbents can be altered chemically or genetically to improve adsorption. In chemical modification, binding sites are either enhanced or they are impregnated on matrixes which enhance adsorption capacity and the selectivity of the adsorbent towards the metal ion [122–125]. The biosorbents shown in Table 6 are capable of selectively adsorbing scandium from low concentration solutions the in presence of low concentration of impurities. When the concentration of impurities increases, the sorbent fails to selectively separate scandium from feed sources, like acid mine drainage or leachate of red mud [123–125]. However, a study by G.I. Karavaiko claimed to selectively recover scandium from diluted red mud leachate using biomass *S. cerevisiae* as the biosorbent [126]. To avoid co-precipitation of scandium along with iron, aluminum and titanium, pH 0.6 was maintained in the system. The biosorbent showed exceptional selectivity towards scandium and after four cycles of adsorption and desorption, 98.9% of scandium was extracted [126]. The process of biosorption has many advantages, such as easy operations, no chemical sludge is produced and it is environmentally friendly. The biosorbents discussed were able to successfully recover scandium from diluted streams with low impurity. However, further work needs to be done on enhancing its selectivity from feeds with high amounts of contaminating metals with similar chemical properties as scandium.

**Table 6.** Summary of different kinds of adsorbent used for recovery of scandium.

| S. No. | Sample | Adsorbent | Resin | Extractant/ Microbe | Adsorption Conditions | Desorption | Removal Percentage (%) | Isotherm Kinetics | Mechanism | Reference |
|---|---|---|---|---|---|---|---|---|---|---|
| 1 | $HNO_3$ leachate of Greek bauxite residue | EGTA-functionalized chitosan–silica | - | - | pH 1.25, Adsorbent dose 25.0 mg, 10.0 mL, initial conc. 0.50 mM, time 4 h | $HNO_3$ at pH 0.50 | 80% | Langmuir | ion exchange | [40] |
| 2 | Sc, Y, La, Ce, Lu, Nd, Sm, Eu, Tb, Dy, Ho, Er, Gd Tm, Yb and Pr | Sol-gel processed silica doped with a novel bifunctional ionic liquid, trioctylmethy-lammonium 1-phenyl-3-methyl-4-benzoylpyrazol-5-onate | - | - | 0.05 M $HNO_3$, V/m = 200 mL/g, 10 min | 2 M $HNO_3$ | - | Langmuir pseudo-second-order | chemisorption | [96] |
| 3 | Model aqueous phase of scandium | $Fe_3O_4@SiO_2$ coupling agent APTES as a and ligand (EDTA) | - | - | initial conc. 50 mg/L, pH 5, 50 mg adsorbent, 5 h, 25 °C | - | 95% | Langmuir pseudo-second order kinetic | exchange or sharing of electrons | [97] |
| 4 | Sc, Fe, Al | - | TP 260 & TP 209 | - | 50 mg resin, 50 mL 1 M $Na_2SO_4$ solution, pH 2 initial conc. 50 mg-Sc/L, 70 °C, 36 h | - | - | Langmuir isotherm | intraparticle diffusion | [98] |
| 5 | Sc, V, Ti | - | Dowex 1 | - | 0.1 M oxalic acid | 0.1 M oxalic acid and 0.1 M HCl | - | - | - | [99] |
| 6 | Th, Zr, Fe, Ti, Al and Ca | - | Diaion SK 1, a styrene-base strong acid type resin | - | 1 mL per min, 10 g of dry resin. | 1 M $NH_4SCN$ and 0.5 M HCl | 100% | - | - | [100] |
| 7 | Yt, La, Ce, Sm, Er, and Yb | - | AG 50W-X8 resin | - | 20 g resin, flow rate of 3.0 mL/min | 2 N sulfuric acid | 100% | - | - | [101] |
| 8 | Sc, Yb, Eu, Ce, Sr, Na, and C | - | cation-exchange resin Dowex 50 | - | 95% $(CH_2)_4O$, 5%, 6 M HCl, 0.1 M TOPO, 1 g resin, flow rate: 0.5 mL/min | - | - | - | - | [102] |
| 9 | Raffinate copper leach solution | - | Purolite C100Na | - | 0.1 g of washed and dried resin, pH-1.5, 25 °C, 24 h | 1.7 M/L $Na_2CO_3$ | - | Langmuir isotherm | - | [105] |
| 10 | REE mixture | - | Dowex 1X4 and Amberlite CG-400, | - | - | 10% 7 M $HNO_3$ (90% methanol) mixture was prepared in 10 mL | scandium was not adsorbed to an appreciable extent | - | - | [106] |
| 11 | Zr-raffinate with REE | - | Anion-exchange resin Dowex 1X8, and cation exchange resins, Dowex 50X8 | - | 0.1 M $HNO_3$ | anion-exchanger in 2.5% 7 M $HNO_3$–$CH_3OH$ mixture cation-exchanger 5% 1.2 M HCl-$(CH_3)_2CO$ mixture, 91% | - | - | - | [107] |

**Table 6.** *Cont.*

| S. No. | Sample | Adsorbent | Resin | Extractant/ Microbe | Adsorption Conditions | Desorption | Removal Percentage (%) | Isotherm Kinetics | Mechanism | Reference |
|---|---|---|---|---|---|---|---|---|---|---|
| 12 | Uranium leachate | - | Tulsion CH 93 | - | 0.1 g sample of air-dried resin 50 mL of solution, shaken, 24 h, 20–23 °C | 180 g/L $Na_2CO_3$, Sc and Th were 94.1 and 98.9%, respectively $(NH_4)_2SO_4$ (50 g/L) a mixture of 30% $(NH_4)_2CO_3$ + 70% $NH_4HCO_3$ (ACBM) | - | - | - | [108] |
| 13 | Red mud | - | AFI-21 and AFI-22 | - | sulfuric acid media pH 0.9–4.9 | NaOH, 20–30 g/L | 50% | - | - | [109] |
| 14 | Sc, Ti, Fe(III), Ca, Al, Zr, Si | - | 732-type acid cation exchange resin | - | pH 2.5, 200 r/min, 0.55 g EDTA and 0.16 g ascorbic acid, pH 2.5, 180 min, 25 °C | - | 84.2% | - | - | [110] |
| 15 | Al, Ti, Fe, Y, La, Ce | - | porous silica-polymer based $TRPO/SiO_2$-P | - | pH 2 $H_2SO_4$ s/L: 1.0 g/50 mL, initial conc. 10 mM, 2 h, room temperature | 0.01 M EDTA | 100 | Langmuir adsorption isotherm | electron sharing | [111] |
| 16 | Sc, Fe, Al | - | TP 272 | Cyanex 272 | 50 mg resin, 50 mL 1 M $Na_2SO_4$ solution at pH 2.5 with initial concentration of 50 mg-Sc/L, 22 °C, 12 h | - | - | Langmuir isotherm pseudo-second-order model | intraparticle diffusion | [113] |
| 17 | Coal, fly ash leachate | - | VP OC 1026, TP 272 | D2EHPA, Cyanex 272 | S/L ratio of 1/100 (wt./vol.) 40 °C 150 rpm, pH 2.33 | 2 M $NH_4F$, 40 °C 6 M $H_2SO_4$, 18 h | 91% 85% | - | adsorbed via proton exchange with the phosphate groups | [114] |
| 18 | La, Dy, Ce, Pr, Nd, Eu, Sm, Gd, Tb, Ho, Er, Yb, Lu, Y, Tm, and Sc | - | Amberchrom CG-71c nonionic macroporous sorbent | P,N-containing podands | 5 M $HClO_4$, ratio of the aq. sol. vol. to the sorbent weight: 100:1 | - | - | - | complexation by enhanced protonation | [115] |
| 19 | $Al^{3+}$, $Fe^{3+}$, $Zr^{4+}$, $Mn^{2+}$, $Co^{2+}$, $Cu^{2+}$, $Ni^{2+}$, and $Zn^{2+}$ | - | XAD-7HP resin | extractants PC-88A and Versatic 10 | 50 mg resin, 5 mL aqueous solution, and 1 h shaking at room temperature | 2 M sulfuric acid | - | Langmuir isotherm and second-order kinetics | - | [115] |
| 20 | Sc, Al, Fe | - | polymer support fabric (PP-g-PGMA) | phenylphosphinic acid (PPI) | 1 ppm Sc, pH 2, 24 h at room temperature | - | 98% | Langmuir | solvation mechanism of adsorption between Sc and PPI | [116] |
| 21 | REE | - | DIAION HP2MG (methacrylate resin) | Cyanex 272 1-octanol as modifier | 20 mL of REE 1 mM solution, 20 mg of SIRs, constant shaking, 24 min, 298 K | 5 M HCl | - | Langmuir | - | [118] |

**Table 6.** *Cont.*

| S. No. | Sample | Adsorbent | Resin | Extractant/ Microbe | Adsorption Conditions | Desorption | Removal Percentage (%) | Isotherm Kinetics | Mechanism | Reference |
|---|---|---|---|---|---|---|---|---|---|---|
| 22 | Sc, Tm, Yb and Lu | - | Modified Merrifield Resins | Cyanex 923, | 5 mL of Sc solution, 200 mg of impregnated resins, constant stirring, 30 min, 25 °C | - | - | - | extraction of neutral complex and cation exchange | [119] |
| 23 | Sc solution | - | TVEX | TBP, di-isooctyl methyl phosphonate (DIOMP) and phosphine oxide with different alkyl groups (POR) | organic to aquas-phase ratio of 1:20, 25 °C, 24 h, >4 M HCl | - | | | TVEX-DIOMP as [ScCl (DIOMP)$_2$ (H$_2$O)$_3$]$^{2-}$ complx from 4 M HCl | [120] |
| 24 | Coal byproduct | - | microbe-encapsulated silica gel (MESG) biosorbent | cell loading of 1.0 g/mL, pH 3.0, 1 bed vol. 2 mL feedstock sol. | pH-6, 0.050 M sodium citrate | - | - | - | - | [122] |
| 25 | Red mud | - | Quaternized Algal/ Polyethyleneimine beads (Q-APEI) with dry algal biomass | pH > 4 SD: 0.6 g/L; 20 °C;40 rpm; 30 h | 0.5 M HCl/CaCl$_2$ solutions 88.1% | - | - | Langmuir | complexation of the Sc with amine groups | [123] |
| 26 | Red mud | - | *Laminaria digitata* algal biomass/ polyethyleneimine beads, A$_L$PEI | pH 1–5, 1 mmol/L. SD: 2 g/L; T: 22 °C, 48 h; 170 rpm | acidic CaCl$_3$, 99% | - | - | Langmuir equation | ion exchange and chelation on protonated amine groups, sulfonic groups and carboxylate groups | [124] |
| 27 | AMD and seawater | - | *Posidonia oceanica* with 1-(2-pyridylazo)-2-naphthol (PAN) grafted on algal biomass (2-algae-P) | Adsorbent dosage 1 g/L. pH 5 (AMD), pH 6 (Seawater), REE = ~2 ppm, 45 °C, 1 h | - | - | - | Langmuir and pseudo-second order kinetics | binding by coordination mechanism with the ligand of PAN | [125] |
| 28 | Red mud | - | - | biomass of *Saccharomyces cerevisiae* and *Aspergillus terreus* | pH 0.6 fungi (0.2 g/L, dry wt) and yeasts (0.5 g/L, dry wt), 20 mL aliquots, 220 rpm, 20–25 °C, 1 h | with 20 mL of 10% *w/v* Na$_2$CO$_3$, 99.5% | *S. cerevisiae* 98.8% | Langmuir equation | - | [126] |
| 29 | Sc, Ce, La and Al (monazite processing liquor) | - | glycol amic acid embedded resin | - | 24 h, pH 1, 0.1 g resin in 100 mL | 2.0 M HCl solution at 80 °C, | 45% | - | - | [127] |
| 30 | Sc and Nu | Biochar of wood dust | - | - | absorbent conc. 1–10 g/L, 24 h, 25 °C, 350 rpm | - | 52% and 78%, respectively | - | - | [128] |

**Table 6.** *Cont.*

| S. No. | Sample | Adsorbent | Resin | Extractant/ Microbe | Adsorption Conditions | Desorption | Removal Percentage (%) | Isotherm Kinetics | Mechanism | Reference |
|---|---|---|---|---|---|---|---|---|---|---|
| 31 | Bauxite residue leachate | $\alpha$-ZrP ($\alpha$-zirconium phosphate) | - | - | 0.05 g of ion exchanger, 20 mL feed solution, pH-1.5, 300 rpm, 18 h, at room temp in HCl media | 2 M/L HCl, two-step elution | 99.9% | Langmuir pseudo-second-order kinetics | chemical reaction at the surface | [129] |
| 32 | Tomtor Deposit leachate | - | Purolite D5041(Phosphorus) and Purolite C115 (carboxyl) | - | volume ratio ion exchanger: solution = 1:300 (for phosphorus) and 1:150 (carboxyl), contact time of 24 h | $(NH_4)HS$ (100 g/L) 2 h, 70–80 °C, 78.9% (phosphorus) 1 M $HNO_3$ solution was able to remove REEs | 99.8–99.9% | - | - | [130] |
| 33 | Red mud | - | AFI-21 ampholyte | - | - | $Na_2CO_3$ conc of 150 g/Ldm$^3$ scandium desorption of 96% | 76% | - | - | [130] |
| 34 | REEs(III) | macro-porous silica based polymer ($SiO_2$-P) based di(2-ethylhexyl) phosphonate adsorbent (HDEHP/ $SiO_2$-P) | - | - | m/V = 0.1 g/ 5 mL, 120 rpm, 30 min, $H_2SO_4$ 5 M | - | - | Langmuir | ion exchange | [131] |
| 35 | Sc, Pd, Pt & Au | carboxymethylchitin (CMCht) hydrogel | - | - | pH 3.9, at initial conc. 100 ppb, adsorbent weight 50 mg, 2 h | - | 35% | - | - | [132] |

## 7. Purification Processes

### 7.1. Nanofiltration

Nanofiltration is a pressure-driven filtration-membrane process. Based on separation by size, nanofiltration is the upper end of reverse osmosis and lower end of ultra-filtration. The general pore size of the membrane varies between 1–10 nanometers. The transport across the membrane occurs due to both fluxes as well as trans-membrane pressure differences. The separation happens by diffusion of the molecules of the solvent through the mass of the membrane, controlled primarily by high trans-membrane pressure. It is a process by which part of the feed passes through semi-permeable membrane. For scandium retrieval, a pre-enrichment of scandium is needed to obtain higher concentrations for the subsequent selective extraction [133]. Here, nanofiltration can offer two vital advantages. First, it offers selectivity and secondly, it may reduce the volume to be extracted downstream, reducing the environmental impact caused by solvent extraction steps [134]. However, a major limitation of nanofiltration is that in high ionic strength solutions, adequate fluxes can only be accomplished via high operational pressures, which surges operational costs. Furthermore, there are limited numbers of commercial membranes that can withstand such highly acidic conditions. Remmen et al. [54] prepared layer-by-layer modified nanofiltration membranes that were optimized with respect to their selectivity towards scandium as well as acid resistance. The synthetic solutions consisted of scandium, iron and HCl. In this solution, under optimized conditions, the membrane retained a maximum of up to 64% scandium, efficiently eliminating iron (the major impurity). In real titanium dioxide pigment wastewater, the proposed membrane showed higher retention of scandium 60% compared to available commercial acid resistant membranes that showed only 50%.

### 7.2. Polymer Inclusion Membranes

Polymer inclusion membranes (PIMs) are membranes made up of a liquid extractant trapped in a polymer-based matrix that is typically prepared with polyvinyl chloride (PVC) or cellulose triacetate (CTA). As a separation technology, PIMs are used as the phase transfer medium, extraction, and pre-concentration unit for several cations and anions. In the past decades, PIMs have been extensively used for hydrometallurgical applications such as extraction and metal recovery because of their high selectivity, stability, durability, etc. [135]. A CTA-based PIM was developed with dioctyl phthalate (DOP) as a plasticizer with binary carrier PC-88A and Versatic 10 (decanoic acid). Scandium was selectively separated from a nitrate solution containing other rare earth metal ions. However, poor elution was an issue, which was addressed by using Versatic 10 along with PC-88A as carrier, thus enhancing the extraction of scandium [136]. A comparative study was done between using the PIM containing the amic acid extractants N-[N,N-di(2-ethylhexyl) aminocarbonylmethyl]glycine (D2EHAG) or N-[N,N-di(2-ethylhexyl)aminocarbonylmethyl]phenylalanine (D2EHAF) as carriers and the commercial carriers 2-thenoyltrifluoroacetone (HTTA) or 2-ethylhexylphosphoric acid mono-2-ethylhexyl ester (PC-88A). It was found that the PIM with the amic acid extractants (D2EHAG) or (D2EHAF) as carriers were more efficient in extracting and stripping scandium from a feed solution containing rare earth metals [137] and base metal ion solution containing iron (III) [138]. It was found that PIM with the amic acid extractant D2EHAF was able to successfully separate scandium in both cases. Thus, the PIM containing D2EHAF was able to quantitatively and selectively transport scandium from 0.1 M sulfate feed solution at pH 3, containing base metal (Fe, Al, Ca, Co, Ni, Mn, Cr and Mg) ions to a solution containing 0.5 M $H_2SO_4$ [138]. However, studies with industrial leach liquor containing scandium have yet to be done.

### 7.3. Precipitation and Crystallization

Precipitation is a separation process where soluble metal ions are separated using other salts by double displacement reactions where the targeted metal ion separates as an in-

soluble solid. The theoretical mechanism is that the dissolved metal ions are precipitated by chemical reagents (precipitants), resulting in the creation of metal hydroxides, carbonates, sulfides, oxalates and phosphates that can be simply separated by filtration, sedimentation or centrifugation. It is mostly used for removal of heavy metals from industrial effluents. Table 7 reports the studies on the recovery of scandium through precipitation and crystallization. The most practiced method of scandium recovery by precipitation is by addition of oxalic acid resulting in scandium oxalate. Calcination of scandium oxalate at around 700 °C produces scandium oxide of 99.5% purity. Apart from oxalic acid, fluorides [65,139] and phosphates have been used for scandium precipitation [140]. However, the main problem is the presence of high concentration of iron (III) which co-precipitates along with scandium. Therefore, for maximum recovery of scandium in pure form, multiple stages of neutralization and precipitation have to be followed to eliminate the major impurities with minimal scandium loss [58,60,141]. Few studies have been reported regarding crystallization of scandium from red mud leachate. Two methods have been used: cooling crystallization and anti-solvent crystallization. The former has a low recovery while the later has a high recovery, but crystal size is very small (< 2 μm), which interferes with separation and filtration. Therefore, better crystallization and filtration techniques must be developed for this process. A uniform distribution of iron in the crystals as an impurity has also been found due to its similarity with scandium with highest purity of 98.3% [142,143]. Thus, the leach liquor feeds that have scandium in the form of $(NH_4)_3ScF_6$ with a low presence of impurities can be converted to crystals of high purity. However, the use of fluoride is of environmental concern. Therefore, further studies in this area are required.

**Table 7.** Summary of precipitation and crystallization methods.

| S. No | Sample | Process | Agent | Condition | Recovery Percentage | Reference |
|---|---|---|---|---|---|---|
| 1 | Uranium leachate | Multiple precipitation | HF<br>Oxalic acid | - | 10% (99.5% purity) | [58] |
| 2 | Scandium and titanium | Neutralization precipitation | $Ca(OH)_2$ | pH-2, 3.3 g/L titanium oxide | 96.75 | [60] |
| 3 | Red mud leachate | Dual-stage successive precipitation | Dibasic phosphate | - | $ScPO_4$ | [65] |
| 4 | Synthetic scandium solution | - | sodium fluoride, ammonium hexafluoroscandate at molar ratios of F to Sc within 1–14 | - | $ScF_3$, $Na_3ScF_6$, and $Na(NH_4)_2ScF_6$ | [139] |
| 5 | $H_2SO_4$ leachate of bauxite | Triple-stage successive precipitation | $NH_4OH$<br>$NH_4OH$<br>$(NH_4)_2HPO_4$ | pH-3.3–3.4<br>pH-3.6–3.7<br>pH-2.5–2.6 | 65% $ScPO_4$ | [140] |
| 6 | Strip liquor containing 0.2 wt% Sc and minute impurities | Cooling crystallization Anti-solvent crystallization | -<br>Ethanol | 1 °C<br>Ethanol-to-strip liquor volumetric ratio of 0.8 | $(NH_4)_3ScF_6$ < 50%<br>98% | [142] |
| 7 | Ammonium scandium hexafluoride from solvent extraction strip liquors | Anti-solvent crystallization | Ethanol | Solvent to anti-solvent volumetric ratio of 1:1 ethanol conc of 8.6 mol/L | 98% purities greater than 98.3% | [143] |
| 8 | Nickle leachate | Neutralization and sulfide precipitation | - | pH > 4 | - | [144] |
| 9 | Tungstenic slag | Extraction | $H_2SO_4$ | - | 94% $ScCl_3$ | [145] |
| 10 | Tungsten slag | precipitation | Oxalic acid | - | 85.2% | [146] |

## 8. Recent Scandium Case Studies

After discussing the unit operations for scandium recovery, we will discuss some recent case studies on scandium recovery (Table 8). There are a number of projects dealing with scandium recovery: the SCALE project, EU; the Nyngan project in New South Wales, Australia; the polymetallic Elk Creek project in Nebraska; the polymetallic Owendale Project in New South Wales; the polymetallic Sunrise Project in New South Wales; the polymetallic SCONI project, Queensland; the Taganito high-pressure acid-leach nickel operation, Japan; byproduct of alumina refining in the Ural Mountains, Russia; and devel-

opment of scandium recovery as a byproduct of uranium production, in Dalur, Kurgan region, Russia.

**Table 8.** Status of projects on scandium recovery.

| S. No. | Country | Name of Project | Primary Resource | Status |
|--------|---------|-----------------|------------------|--------|
| 1. | Australia | Nyngan scandium project | Typical tertiary laterite composed of limonites and saprolites | The feasibility study concludes that the project has the potential to produce an average of 37,690 kg of scandium oxide per year, at grades of 98.0–99.9% |
| 2. | Nebraska, US | EIK Creek Niobium project | Carbonatite rocks | The mine is expected to produce 168,861 t of niobium in the form of ferroniobium, 3410 t of scandium oxide and 415,841 t of titanium dioxide over its operating life of 36 years |
| 3. | New South Wales, Australia | Owendale scandium project | Platina resources | Stage one will produce 20 tons per annum (tpa) of scandium oxide during the initial five years of operation, while stage two will double the annual production capacity to 40 t with the processing plant upgrade |

## 9. Environmental Risk Assessment

Extraction of REEs from their primary sources has led to severe environmental hazards. Presently, scandium is mainly recovered as a byproduct from the processing of other metals such as titanium dioxide, REEs and aluminum [12]. This makes the retrieval of scandium technically challenging [12] and environmentally hazardous [91]. The Mountain Pass mine in California, which controlled the global REE market for decades, was closed down in 2002 because of the pollution and environmental hazard it generated [146–148]. Similarly, the REE sources in China have now raised serious health and environmental concerns, such as emission of hazardous gases like $H_2S$, HF and $H_2SO_4$ [149]. It also causes heavy metal leakage to land groundwater as well as exposure to radioactive metals around the mining sites [150]. Moreover, prolonged mining leaves fine dust particles suspended in the air, long time inhalation of which gives rise to pneumoconiosis (black lung) [151]. The City of Baotou, located near the Bayan Obo mine, had many pollution issues caused by mining and nearby processing of REEs. The Chinese government has therefore closed many facilities as well as limited production based on the environmental damage [152].

LCA is a source-to-grave and, sometimes, especially with recycling units, source-to-source investigation techniques to evaluate environmental impacts accompanying all the stages of a product's life from raw materials to its distribution or recycling. It can assist decision makers to implement the means needed to improve the sustainability performance of the unit [152]. In short, LCA provides a complete outlook of the environmental effects caused over the entire life cycle of the product. It starts from raw material extraction and attainment, followed by manufacturing, transportation, supply, maintenance, reuse and recycling, and disposal and waste management [153]. Major features of LCA are that it avoids shifting environmental consequences from one geographical area, source and stage to another. The four main stages of LCA are shown below in Figure 2 that depicts the stages followed for life cycle assessment of any product. The first stage of LCA analysis defines what part of the product's life cycle will be used for analysis, thus defining the boundary for the assessment. The second stage provides information about the mass and energy flow across the set boundary in relation to raw materials consumed in these processes. In this stage, details about the environmental impact in terms of energy consumption, emission and interactions are also given. It serves as an inventory for impact assessment in the next stage. In the third stage, the inventory formed in the previous stage is analyzed in detail. Results in terms of indicators such as Human toxicity non-cancer (HTNC), Global Warming Air (GWA), Eutrophication, Toxicity and others are studied in detail for all impact categories. Every impact category is assessed in detail by weighing and normalizing in the third stage. Once the assessment is over, interpretation of the assessment is done in the fourth and final stage of LCA. It involves critically reviewing

the data, understanding the sensitivity of the assessment and representation of the results. Thus, with LCA, a map can be drawn between the environment burdens such as pollutants and waste generated after each stage and the consumption of energy and raw material for those stages individually. This will help in assessing the effect of the input and output parameters associated with the particular product and the accompanying pathway for the long-term sustainability of energy sources, human health, climate, biodiversity and many more. Thus, LCA helps in identifying the most suitable pathway for a particular product with the lowest environment burden. It generates a scope to identify the steps and their alternative approaches to reduce the negative impacts on the environment. The critical and specific nature of REEs, including scandium, demands an LCA to provide proper guidance for industries to adopt more eco-friendly approaches for downstream of the REEs. Many LCA studies have been done for accessing the environmental effects of REEs [152,154–156]. However, the environmental effects of scandium have been specifically studied only by Wang et al. [27]. From their study, it was concluded that HTNC and GWA are the top two significant challenges of $Sc_2O_3$ production. Steam and oxalic acid are the top two inputs with the greatest impacts on HTNC and GWA caused indirectly during production stage. Thus, reduction in the consumption of the above two inputs could reduce the overall impact associated with scandium mining. Moreover, use of alternative reagents with lower environmental impact or reduction of the dosage could be done to reduce the impact, were some of the suggestions provided. As scandium is predominantly recovered from bauxite residues, LCA can provide a better outlook towards its environmental effect. A recent LCA study was done by Joyce and Björklund [27] to understand the environmental effect of recovering valuable metals from bauxite residue with the primary goal of "Zero-Waste Valorization". Few observations were made by the authors, one of them was that the merging of technologies proved to be beneficial. For instance, combination of valorization with pretreatment can lead to environmental benefits. Moreover, modification and utilization of pathway residues for other purpose like construction would also decrease environmental impacts. Another important parameter pointed out was the incorporation of recycle and re-use of pathway reagents [156]. Thus, the key of LCA analysis is the acknowledgement that simple parameters of the pathway that may seem environmentally harmless in isolation can have major environmental impact indirectly when used in the processing stages. Therefore, this analysis gives us a close understanding of the process, different parameters and their effects on the environment.

However, since LCA analysis cannot be transferred to other sources or geographical locations, it is very important to assess the environmental effect of other primary and secondary sources as well. Moreover, process pathways associated with recycling and reuse must be assessed to understand the effect on the ecology. The recycling source pathways have many added advantages over virgin primary sources such as elimination of a number of processes, which has been seen to significantly contribute to the environmental impacts like ore mining, leaching etc. It also avoids undesirable byproducts which are fundamental to mining from primary sources [157]. Despite all the benefits associated with recycling as a source, it has not been explored properly for LCA to understand the environmental impacts. As the burden of mining has caused enough environmental damage to date, it is important to at least develop a Life Cycle Thinking (LCT) approach, with the initial source being recycling of valuable and critical metals. It will provide better understanding of the pathways and even give scope for the development of processes with a focus on environmental benefits. This will help decision and policy makers to make the appropriate decisions. It will further help to shift the burden of mining of major metal ores to valuable and critical metals, improve the associated global economic market of these metals, reduce the accompanying environmental effects, and provide a scope for more technological advancements in that arena.

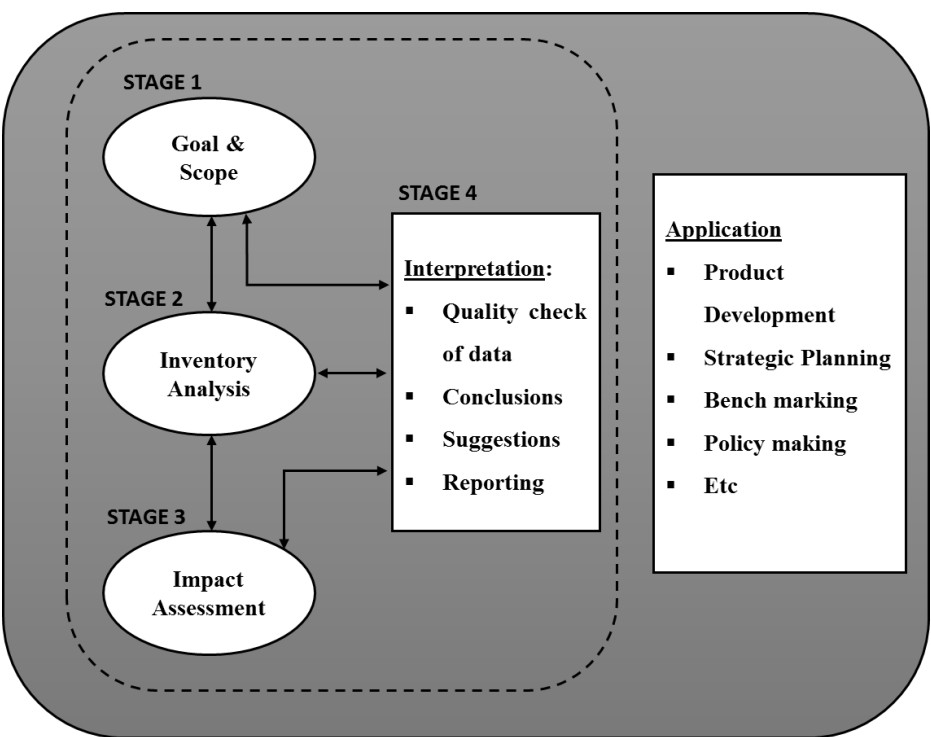

**Figure 2.** The stages of LCA analysis.

## 10. Conclusions

New technologies involving the use of scandium is emerging with time. Over the decades, the two main applications of scandium were in solid oxide fuel cells and aluminum–scandium alloys. However, due to its unique physico-chemical properties, scandium has sufficiently heightened its market demand with new applications in future aircraft and automotive manufacturing. Globally, scandium is found across many nations in the ores of other metals such as aluminum, uranium, tantalum and others. However, the largest contributor of scandium is the reserve of Nb–REE–Fe in Bayan Obo, Inner Mongolia, China, where it is generated from the REEs tailings as depicted in Figure 1. In addition, scandium is largely recovered from aluminum ore (bauxite) processing waste as a secondary source. However, due to the presence of other metals with similar geochemistry as scandium, efficient recovery of the metal becomes quite challenging. Many other sources have been identified but have not been explored, like electrical and municipal waste. Hence, exploration of urban mining as a viable source for scandium recovery is needed. Commercial recovery of scandium takes place following a number of different types of hydrometallurgical processes. These hydrometallurgical processes add to the environmental burden in the form of high energy consumption, utilization of toxic chemicals, production of secondary toxic waste, etc. Hence, this review outlines the technological advances over the decades across various hydrometallurgical separation processes stating their pros and cons as well as advantages of one process over another. An attempt has been made to identify some suitable advances based on their reduction in environmental damage and high scandium recovery efficiency. Since there are few discussions of the environmental impact of the technical advances in literature, it is hard to analyze and state their overall impact on the environment when applied for commercial recovery. As reported and discussed, there has been very limited studies on the environmental impact of commercial scandium recovery but they clearly shows that, the beneficiation and hydrometallurgy steps in the process contributes major burdens on the environment primarily because of low concentrations of scandium and the presence of high concentrations and numbers of impurities. Hence, it is important to identify and explore alternate sources as well as alternate hydrometallurgical processes that can lead to the reduction of the burden and impact on the

environment. Moreover, there is a huge gap and lack of attention towards the analysis of the environmental impact of scandium production from different available sources following different pathways. This could help in getting a clearer comparison and understanding of the relationship between the processes and the environmental burdens generated by them so that further studies could be done to reduce the environmental burden.

## 11. Understanding and Future Direction

The review presented here has been prepared with a goal of giving readers an understanding of the roots of the environmental burden caused by scandium production. There are few studies that discuss the environmental impact of scandium recovery from different sources. The authors have tried to summarize that information that is available. It is very well understood that scandium is associated with REEs and other metals when it is recovered from primary, secondary or tertiary sources; and, since the percentage of scandium is very low compared to other metals, its recovery has become expensive, challenging and environmentally hazardous. Thus, the removal of other metals along with efficient recovery of scandium is most important parameter that can help in reduction of environmental burden. There have been many technological advances. The authors have outlined the advancements made over years in different areas of technology with a focus on the enhancement of recovery efficiency of scandium. Each process discussed has been outlined with the associated benefits and limitations for the readers to understand and take up research in the unexplored areas that can benefit both the economy and the environment. For example, few studies have been done on the recovery of scandium from electronic waste. This source can be explored, as it will promote recycling as well as reduce the mining burden to some extent. Moreover, more exploration of bioleaching, biosorption, solid extractants and other methods with the aim of high selectivity for scandium needs to be done. Additionally, the unavailability of many studies on the life cycle assessment of scandium from different types of sources, very well reflects the lack of environmental awareness linked with scandium recovery. Therefore, more research needs to be done in this area to truly understand the role of the source and the processing pathway followed for scandium recovery. Thus, we will have a better idea of the direction to work in to achieve the goal of eco-friendly processes and sustainability.

**Author Contributions:** All authors contributed to the study conception and design. Material preparation and data collection were performed by A.G. (Aratrika Ghosh), S.D., A.G. (Anirudh Gupta) and R.J. The first draft of the manuscript was written by A.G. (Aratrika Ghosh), S.D. and A.G. (Anirudh Gupta) and all authors commented on previous versions of the manuscript. All authors have read and agreed to the published version of the manuscript.

**Funding:** The authors declare the support of the IGSTC project "BioCuInGe" with sanction No.: IGSTC/CALL2016/BIO-CULNGE/132017.

**Informed Consent Statement:** Not applicable.

**Data Availability Statement:** All data are true and valid and are available.

**Acknowledgments:** The authors would like to thank the IIT Delhi and HZDR administration for providing the necessary facilities to carry out the present study.

**Conflicts of Interest:** The authors declare no conflict of interest.

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
