# Peer review of "Process Evaluation of Scandium Production and Its Environmental Impact"

_environments, doi:10.3390/environments10010008_

Round 1

Author Response

All the comments of the reviewer 1 were addressed

Reviewer 2 Report

This manuscript summarized the new technologies for scandium application and recovery. The authors investigated the scandium sources, application, separation, and recovery in detail. However, this manuscript did not include one important Sc source, coal and coal by products (CCP). I recommend including CCP in this review paper. Below are some important references for Sc recovery from CCP:

1.     Taggart, R. K., Hower, J. C., Dwyer, G. S., & Hsu-Kim, H. (2016). Trends in the rare earth element content of US-based coal combustion fly ashes. Environmental science & technology50(11), 5919-5926.

2.     Zhang, W., & Honaker, R. (2020). Characterization and recovery of rare earth elements and other critical metals (Co, Cr, Li, Mn, Sr, and V) from the calcination products of a coal refuse sample. Fuel267, 117236.

3.     Hussain, R., & Luo, K. (2020). Geochemical evaluation of enrichment of rare-earth and critical elements in coal wastes from Jurassic and Permo-Carboniferous coals in Ordos Basin, China. Natural Resources Research29(3), 1731-1754.

4.     Lefticariu, L., Klitzing, K. L., & Kolker, A. (2020). Rare Earth Elements and Yttrium (REY) in coal mine drainage from the Illinois Basin, USA. International Journal of Coal Geology, 217, 103327.

5.     Kermer, R., Hedrich, S., Bellenberg, S., Brett, B., Schrader, D., Schoenherr, P., ... & Janneck, E. (2017). Lignite ash: Waste material or potential resource-Investigation of metal recovery and utilization options. Hydrometallurgy, 168, 141-152.

6.     Honaker, R. Q., Groppo, J., Yoon, R. H., Luttrell, G. H., Noble, A., & Herbst, J. (2017). Process evaluation and flowsheet development for the recovery of rare earth elements from coal and associated byproducts. Minerals & Metallurgical Processing, 34(3), 107-115.

7.     Honaker, R. Q., Zhang, W., & Werner, J. (2019). Acid leaching of rare earth elements from coal and coal ash: implications for using fluidized bed combustion to assist in the recovery of critical materials. Energy & Fuels33(7), 5971-5980.

8.     Dong, Z., Deblonde, G., Middleton, A., Hu, D., Dohnalkova, A., Kovarik, L., ... & Park, D. (2021). Microbe-encapsulated silica gel biosorbents for selective extraction of scandium from coal byproducts. Environmental Science & Technology, 55(9), 6320-6328.

Author Response

All the comments by the reviewer 2 have been addressed and references suggested by him/her have been added.

Reviewer 3 Report

1) The authors described the theme of the present is related to the process evaluation for Sc production and its environmental impact; however, this reviewer could not find out much on the environmental aspects which should be the actually emphasized in the review as per the scope of this journal. Hence, it is recommended to revise the manuscript accordingly and provide the adequate information on its environmental aspects. Only a figure on LCA does not justify the theme of the manuscript that too without having a clarity on that. Can the authors tell for what Fig. 1 stands of and using the defined factors what information and how it can be drawn?

2) After going through the manuscript, this reviewer could not find out any relevance of section 4. Not a single flow sheet has been discussed in the manuscript. I would like to suggest to put some commercial flow sheets on Sc extraction and recovery and then the environmental impacts of those flow sheets can be discussed.

3) In section 5: why the authors discussed a 5 years old market price of Sc? It shows that the authors did not pay attention to their review otherwise finding the recent metal price is not a difficult task.

4) The manuscript did not discuss any high-temperature process. Does it present the limitation of this manuscript? if yes, then add this limitation in the introduction of the manuscript.

5) For me, it seems like a bunch of published articles information gather at one place, which is actually not hard to do using the online available materials.

6) There is no description of the 2nd illustration of the manuscript.

7) The most of the sentences in conclusions actually do not reflect the concluding remarks instead they are general statements. Better to re-write the conclusion with a focus on the topic and actual conclusion that can be drawn from this review.

8) A throughout analysis of the manuscript did not give any idea of the authors on the reviewed topic. A separate section on the authors understanding of the topic, their suggestions, and future directions should be added in the manuscript.

9) Overall, the manuscript has several weakness, making it not suitable for publication at its current version.  

Author Response

All the comments by the reviewer three have been addressed. In particular, sustainability aspect of Sc process as well as authors understanding on the Sc process and future direction has been added. Further, the conclusion section has been completely rewritten. Further, the reasons for not reviewing pyrometallurgical processes has also been explained in the manuscript. 

Round 2

Reviewer 1 Report

The chemical formula H2SO4 (item 5 in Table 7) should be corrected.

Author Response

The reply has been added in the uploaded file

Reviewer 3 Report

The authors have significantly revised the manuscript, reaching up to the acceptance level.

Author Response

We thank the reviewer 3 for his comments. 
